# Evasion of the Antiviral Innate Immunity by PRV

**DOI:** 10.3390/ijms252313140

**Published:** 2024-12-06

**Authors:** Chenlong Wang, Longxi Li, Xinyu Zhai, Hongtao Chang, Huimin Liu

**Affiliations:** 1College of Life Sciences, Henan Agricultural University, Zhengzhou 450046, China; wangchenlong1230@163.com (C.W.); llx13353809429@163.com (L.L.); zhaixinyu2020@163.com (X.Z.); 2College of Veterinary Medicine, Henan Agricultural University, Zhengzhou 450046, China

**Keywords:** PRV, innate immunity, immune evasion

## Abstract

Pseudorabies virus (PRV) establishes persistent latent infections by effectively evading the host’s antiviral innate immune response. PRV has developed sophisticated strategies to bypass immune surveillance through coevolution with its host. Currently, no effective vaccine exists to prevent or treat infections caused by emerging PRV variants, and the interactions between PRV and the host’s innate immune defenses remain incompletely understood. Nevertheless, ongoing research is uncovering insights that may lead to novel treatments and preventive approaches for herpesvirus-related diseases. This review summarizes recent advances in understanding how PRV disrupts key adaptors in immune signaling pathways to evade antiviral immunity. Additionally, we explored the intrinsic cellular defenses that play crucial roles in combating viral invasion. A deeper understanding of the immune evasion strategies of PRV could inform the development of new therapeutic targets and vaccines.

## 1. Introduction

Pseudorabies (PR), also known as Aujeszky’s disease, is a highly contagious disease caused by the pseudorabies virus (PRV). PRV belongs to the alpha-herpesvirus subfamily and is an enveloped virus with a large linear double-stranded DNA genome encoding more than 70 proteins [1]. While pigs are natural hosts, PRV can infect a wide range of mammals [2]. Although vaccination strategies have been well performed in North America and Europe, PRV remains a significant swine disease in many regions, including China. The emergence of PRV variants in China in 2011 caused significant economic losses in the pig industry, complicating eradication efforts [3,4]. Long-term immune pressure may contribute to PRV variation, enabling immune evasion and creating new challenges for future control efforts. Recently, reports on human cases of PRV infection further revealed the potential risk of PRV transmission from pigs to humans, which poses an increasing threat to public health [5,6,7]. 

Like other alpha-herpesviruses, PRV can establish latent infections in its natural hosts and is often used as a model virus to study the biology of alpha-herpesviruses and host innate immune responses. The PRV genome is complex and highly structured, consisting of unique long (UL) and unique short (US) regions flanked by inverted repeat sequences. These regions encode proteins responsible for viral replication, virulence, and immune evasion. The PRV life cycle begins with the virus attaching to host cells through the viral glycoproteins gD, gC, gB, gH, and gL. Following attachment, the virus fuses with the host’s plasma membrane, allowing entry into the cell. The viral capsid and associated proteins are then transported to the nucleus, where the viral DNA is released. Inside the nucleus, the virus undergoes sequential gene expression, starting with the immediate-early gene IE180, followed by early genes involved in DNA replication and, finally, late genes encoding structural proteins. Newly synthesized viral DNA is packaged into capsids, which are assembled in the nucleus. The mature virions are enveloped and released from the cell to infect new cells (Figure 1). PRV can also establish latency in sensory neurons, remaining dormant until sensory neurons reactivate under certain conditions [1,8].

Viral infections trigger rapid immune responses from the host to combat the infection. Among these responses, the type I interferon (IFN) pathway plays a pivotal role in viral suppression, immunomodulation, and adaptive immune response regulation [9,10]. Upon viral infection, host cells activate multiple signaling cascades that recognize specific vial components, disrupting viral replication. The recognition of viral DNA and the activation of downstream signaling pathways are critical in antiviral innate immunity. Cytosolic DNA sensors, such as Toll-like receptors (TLRs), RIG-I-like receptors (RLRs), IFI16 (interferon-gamma-inducible protein 16), AIM2 (absent in melanoma 2), and cGAS (cyclic GMP-AMP synthase), are essential for detecting DNA viruses such as PRV [11,12,13,14,15]. Once detected, these sensors initiate cellular signaling cascades that lead to the production of IFNs, the expression of IFN-stimulated genes (ISGs), and the establishment of an antiviral state [16,17].

As a DNA virus, PRV can establish persistent infections by evading IFN-I-mediated innate immune responses. This review examines recent findings on how PRV circumvents IFN-I-mediated antiviral immunity and the associated downstream signaling pathways [18].

## 2. PRV Evades Pathogen Recognition Receptor-Mediated Antiviral Innate Immunity

The expression of pathogen recognition receptors (PRRs) equips host cells with the ability to detect and respond to viral infections. Upon recognizing viral ligands, TLRs and other cytosolic receptors activate downstream adaptor proteins, triggering signaling cascades that lead to the production of IFN-I and the subsequent activation of ISGs. However, to ensure survival and replication within host cells, viruses, including PRV, have evolved sophisticated strategies to counteract the host’s antiviral innate immunity. This section discusses the classical innate immune responses initiated by PRRs, including TLRs, RLRs, DNA sensors, nucleotide-binding oligomerization domain-like receptor (NLR) sensors, and the downstream IFNAR–JAK–STAT signaling pathway. We next explore the countermeasures of PRV, which are designed to evade or suppress these immune responses, with a focus on how the virus strategically restrains the host’s antiviral defenses to promote its own survival and replication.

### 2.1. TLR Signaling Pathway

The Toll-like receptor (TLR) signaling pathway plays a central role in the innate immune system, functioning as an essential defense against viral infections [19]. TLRs detect pathogen-associated molecular patterns (PAMPs) and initiate signaling cascades that culminate in the production of cytokines, IFNs, and other antiviral molecules. Surface TLRs, such as TLR2 and TLR4, recognize viral envelope proteins, whereas intracellular TLRs, such as TLR3 and TLR7/TLR8, detect viral nucleic acids within endosomes [20,21,22]. Additionally, TLR9 recognizes viral DNA, specifically detecting unmethylated CpG motifs often found in the DNA of viruses [23]. Once these TLRs are engaged with their respective ligands, downstream signaling is initiated through the Toll/interleukin-1 receptor (TIR) domain [24,25], triggering the recruitment of adaptor proteins, such as MyD88 or TRIF, which in turn activate transcription factors, such as NF–κB, IRF3, and IRF7. The activation of these transcription factors leads to the production of IFNs, proinflammatory cytokines like IL-1β, and other antiviral molecules crucial for host defense against viral infection [22,26,27].

Viruses have developed mechanisms to interfere with TLR signaling pathways to evade immune detection and suppress antiviral responses. Herpesviruses, such as HSV, HCMV, and CMV, are known to produce proteins that target TLRs or downstream adaptor proteins, such as MyD88 and TRIF, disrupting the innate immune response [28,29,30,31,32,33]. Similarly, PRV has evolved strategies to evade TLR-mediated immune responses, allowing it to suppress the host’s innate immune responses. Studies suggest that TLR2, TLR3, TLR4, TLR5, and MyD88 signaling are critical for upregulating pro-IL-1β and driving the inflammatory response during PRV infection. TLR5 has been shown to play a key role in inducing the expression of IL-1β, indicating a more significant role for this receptor in PRV-mediated immune responses than previously understood. The activation of TLR2 and TLR4 may depend on viral envelope proteins, further linking these receptors to PRV infection [34]. A study by Ye et al. revealed that the cathelicidin CATH-B1 activates type I IFNs via a TLR4/JNK/IRF3-dependent pathway, highlighting a novel mechanism by which the host can inhibit PRV infection and regulate antiviral responses [35] (Figure 2). These evasion tactics help the virus replicate and persist in the host, underscoring the complexity of the interactions of PRV with the innate immune system. Further research is needed to fully elucidate the precise mechanism by which PRV targets TLR signaling pathways to evade immune detection.

### 2.2. RLR Signaling Pathway

The RIG-I-like receptor (RLR) signaling pathway plays a critical role in detecting viral infections, primarily through the recognition of RNA virus signals. RLRs include retinoic acid-inducible gene-I (RIG-I), melanoma-differentiation-associated gene 5 (MDA5), and laboratory of genetics and physiology 2 (LGP2), which are key intracellular pattern recognition receptors (PRRs) [36,37]. Interestingly, researchers have discovered that even DNA viruses can produce RNA intermediates during replication, which can act as ligands for RLRs [38,39,40,41]. Additionally, virus-derived RNAs generated by DNA viruses can also trigger the RLR-mediated signaling pathway [42]. These findings highlight the importance of RLR-mediated antiviral innate immunity in the defense against DNA viruses. In response, viruses have evolved various strategies to evade this immune mechanism [39].

In alpha-herpesviruses, tegument proteins are major antagonists of RLR signal transduction [43,44]. For PRV, the tegument protein UL24 has been shown to impair RIG-I signaling, leading to the reduced transcription of IFN and ISGs. Moreover, UL24 inhibits RIG-I-induced expression of endogenous oligoadenylate synthetase-like (OASL) proteins in an IRF3-dependent manner, counteracting the antiviral effects of OASL [45]. Recently, Zhao et al. reported that the PRV tegument protein UL13 serves as an antagonist of RLR-mediated antiviral responses by inhibiting the transcription of RIG-I and MDA5 but not LGP2. UL13 achieves this by suppressing NF–κB activation, which in turn reduces RLR-mediated production of IFN-β [46] (Figure 2). These findings reveal a novel immune evasion strategy employed by PRV, although further research is needed to understand these mechanisms fully.

### 2.3. NLR Signaling Pathway

The NLR signaling pathway is crucial for host defense, primarily by activating the inflammasome in response to viral infections. Upon activation, inflammasomes promote the maturation and secretion of proinflammatory cytokines, which are essential for initiating inflammatory responses [39,47]. Among these, the NLRP3 inflammasome is the most extensively studied and can be triggered by PAMPs, damage-associated molecular patterns (DAMPs), and environmental irritants such as viruses, reactive oxygen-free radicals (ORSs), and ATP. Once activated, the NLRP3 inflammasome induces the release of the proinflammatory cytokines IL-1β and IL-18, both of which are essential in antiviral innate immunity [48]. Research has shown that PRV infection primes peripheral nervous system (PNS) neurons for an inflammatory response regulated by TLR2 and IFN-I signaling, leading to a lethal inflammatory outcome in both in vitro and in vivo models [49]. Moreover, Yet et al. demonstrated that PRV infection triggers NLRP3-mediated inflammatory responses, significantly increasing pathogenicity in PRV-infected mice by modulating the host immune response [50]. Thus, the role of the NLRP3 inflammasome during PRV infection is multifaceted and requires further investigation.

The NLRP3 inflammasome is activated to release IL-1β and induce inflammation, while another cytosolic receptor, AIM2, also participates in the release of IL-1β by recognizing viral DNA [51,52]. Zhou et al. reported that PRV infection activates the AIM2 inflammasome both in vitro and in vivo, leading to the maturation and secretion of IL-1β. Mechanistically, PRV infection enhances the expression of pro-IL-1β and pro-IL-18 through the TLR2–TLR3–TLR4–TLR5–NF–κB axis. Interestingly, treatment of PRV-infected peritoneal macrophages with the NLRP3 inhibitor MCC950 did not affect IL-1β secretion, suggesting that NRP3 may not be involved in PRV-induced IL-1β production [34]. Sun et al. further demonstrated that PRV infection upregulated the expression of NLRP3, procaspase-1, GSDMD, pro-IL-1β, and pro-IL-18, resulting in the activation of NLRP3-mediated inflammatory responses and pyroptosis through GSDMD cleavage [51].

### 2.4. DNA-Sensing Signaling Pathway

The DNA-sensing signaling pathway plays a vital role in detecting and responding to viral infections. TLR9, the first DNA sensor identified in 2000 [11], was followed by the discovery of several additional sensors, including cGAS, STING, DDX41, DHX9, IFI16, and ZBP1 [53,54]. Among these, cGAS stands out as the only universal cytoplasmic DNA sensor [15]. Upon recognizing PAMPs, cGAS undergoes conformational changes, leading to the production of cGAMP and the subsequent activation of STING. Once activated, STING, located in the endoplasmic reticulum (ER), rapidly dimerizes and is transported via COPII-dependent anterograde transport through the ER–Golgi pathway. In the Golgi, STING undergoes palmitoylation and recruits TBK1 for phosphorylation. This results in the phosphorylation of IRF3, which dimerizes and translocates to the nucleus to activate IFN-I. Concurrently, TBK1 and its homolog IKKβ enter the nucleus to activate NF–κB signaling, promoting the production of inflammatory cytokines and increasing antiviral immune responses [55,56,57].

The DNA-sensing pathway activated by innate immunity plays a crucial role in controlling PRV infection. However, PRV employs various mechanisms to antagonize this pathway. For example, stimulating cellular DNA-sensing pathways by inducing genomic DNA damage or reducing ectonucleotide pyrophosphatase/phosphodiesterase 1 (ENPP1) levels, which increase cellular cGAS levels, can help mitigate PRV infections [58,59]. While it remains unclear whether PRV has evolved specific mechanisms to evade viral detection, studies have shown that PRV dampens the STING signaling pathway through viral proteins, such as UL13, UL24, and gE/gI. Despite the essential role of the DNA sensing-mediated innate immune pathway in limiting PRV infection, numerous viral proteins counteract immune responses by targeting cGAS, STING, IFI16, and downstream signaling adaptors such as the TBK1–IRF3 axis and NF–κB.

#### 2.4.1. cGAS

cGAS synthesizes cGAMP as a second messenger to activate STING, establishing a critical pathway for detecting nonself DNA and initiating an effective immune response [60]. However, PRV likely employs multiple strategies to disrupt the cGAS–STING pathway. Several studies have suggested that various PRV-encoded tegument proteins regulate antiviral innate immunity mediated via cGAS–STING signaling, facilitating viral replication and latent infection [18]. For example, PRV UL13 has been shown to hinder IFN-β production by directly affecting IRF3 in a kinase activity-dependent manner [61,62]. Additionally, PRV UL24 efficiently inhibits IFN production through the cGAS–STING pathway by interacting with interferon regulatory factor 7 (IRF7) and degrading its expression [63].

Recent research has shown that the PRV tegument protein UL21 binds to cGAS and degrades cGAS via the autophagy–lysosome pathway. Mechanistically, UL21 scaffolds the E3 ligase UBE3C (ubiquitin protein ligase E3C) to catalyze the K27-linked ubiquitination of cGAS at Lys384, which is then recognized by the cargo receptor TOLLIP and degraded in the lysosome. In vivo studies have also demonstrated the role of UL21 in degrading cGAS to promote viral infection [64]. This process of ubiquitination and autophagic degradation of cGAS mediated by UL21 plays a crucial role in the immune evasion of herpesviruses.

Furthermore, the PRV US3 protein has been shown to inhibit IFN production through the cGAS–STING pathway by interacting with IRF3 and promoting its degradation [65]. Yan et al. also reported that the PRV UL38 protein suppresses cGAS–STING-induced antiviral signaling by degrading STING via the autophagy pathway, with UL38 facilitating selective autophagy through TOLLIP [66]. Additionally, PRV gE has been shown to inhibit cGAS–STING-induced IFN production by targeting and degrading CREB-binding protein (CBP) [67] (Figure 3).

#### 2.4.2. STING

Upon sensing dsDNA, cGAS synthesizes the second messenger, cGAMP, which binds to STING, initiating a cascade of antiviral signaling. STING activation is controlled by several mechanisms, including its dimerization, translocation from the endoplasmic reticulum (ER) to the Golgi apparatus and perinuclear region, and post-translational modifications such as phosphorylation and ubiquitination [68,69]. Studies have shown that PRV UL13 targets both STING and IRF3, inhibiting STING-mediated antiviral signaling pathways [18,61,70].

A recent study revealed that the PRV UL38 protein promotes STING degradation as a strategy to counteract the host immune response. Mechanistically, UL38 engages the autophagic receptor TOLLIP, which identifies STING and facilitates its degradation via the lysosome. This degradation inhibits STING activation, suppresses type I interferon production, and enhances viral replication efficiency [66]. Additionally, PRV US2 has been shown to downregulate the antiviral immune response by targeting STING, being involved in viral immune evasion [71]. Kong et al. further found that PRV US2 interacts with STING and recruits the E3 ligase TRIM21, which inhibits IFN signaling and aids in evading the host antiviral defenses, highlighting a novel strategy by which PRV circumvents immune responses [71].

#### 2.4.3. IFI16

IFN-γ inducible protein 16 (IFI16) is a DNA sensor that plays a critical role in antiviral immunity by activating the STING/TBK1/IRF3 signaling pathway [72]. Unlike other DNA sensors, IFI16 can detect both RNA and DNA in the nucleus and cytoplasm [73,74]. Studies suggest that IFI16 plays a crucial role in antiviral immunity against HSV-1 [75], HCMV [76], KSHV [77], and HIV [78]. Upon recognizing viral DNA or RNA, IFI16 activates STING-dependent signaling via the TBK1–IRF3 or IKK–NF–κB axis, leading to the production of type I IFNs, proinflammatory cytokines, and chemokines to mount an antiviral response. However, the role of IFI16 in PRV infection has been less explored. Zhang et al. recently reported that PRV infection activates the IFI16 inflammasome and that both the NLRP3 and IFI16 inflammasomes are involved in pyroptosis during PRV infection. Cleaved GSDMD, activated caspase-1, elevated IFI16 levels, and increased NLRP3 levels were observed in PRV-infected tissues (brain and lung), providing strong evidence of pyroptosis and inflammasome activation in PRV-infected pigs [79].

### 2.5. TBK1–IRF3

The TBK1–IRF3 axis is central to the antiviral innate immune response. Upon binding of cGAMP to STING, STING undergoes conformational changes that recruit TBK1 and IRF3 to the complex. TBK1, a serine/threonine kinase, phosphorylates IRF3, which then translocates to the nucleus to initiate type I IFN production [80]. Since the TBK1–IRF3 axis is essential for antiviral defense, PRV has evolved mechanisms to block this pathway. Lv et al. discovered that peroxiredoxin 1 (PRDX1), an antioxidant enzyme, enhances TBK1/IKKε-mediated IFN-β signaling, thus inhibiting PRV propagation. However, PRV UL13 interacts with PRDX1, promoting its degradation via K48-linked ubiquitination and effectively suppressing the host immune response [81]. PRV UL13 also inhibits type I IFN production by promoting the ubiquitination and degradation of IRF3 [62]. Bo et al. further demonstrated that although PRV UL13 does not affect IRF3 dimer formation, nuclear translocation, or interaction with the CBP coactivator, it markedly reduced the ability of IRF3 to bind to IRF3-responsive promoters [61].

### 2.6. NF–κB

The NF–κB signaling pathway plays a critical role in orchestrating innate immune responses that control infections [82,83]. Viruses, including PRV, often manipulate the NF–κB pathway to facilitate their proliferation and evade immune detection [84]. The PRV-encoded immediate early protein ICP0 inhibits the TNF-α-mediated NF–κB signaling pathway [85]. Additionally, PRV UL24 significantly degrades p65, a key NF–κB component, through the proteasome pathway, facilitating viral evasion of the NF–κB response [86]. Zhao et al. identified a novel evasion strategy in which PRV UL13 suppresses RIG-I and MDA5 transcription by inhibiting NF–κB activation, further highlighting UL13’s pivotal role in modulating host antiviral immune responses [46]. Another study revealed that PRV infection activates the DNA damage response (DDR), which in turn persistently activates NF–κB, potentially explaining why PRV strongly suppresses NF–κB-dependent gene expression [87]. These findings suggest that the interaction of PRV with the DDR–NF–κB axis reflects viral subversion of the NF–κB pathway, which is essential for early host defense. The activation of NF–κB may, therefore, represent a significant barrier to efficient viral replication and dissemination in vivo, but PRV, like other alpha-herpesviruses, effectively circumvents NF–κB-dependent gene expression in infected cells.

## 3. The IFNAR–JAK–STAT Signaling Pathway and Its Downstream ISGs

The IFNAR–JAK–STAT signaling pathway plays a central role in antiviral innate immunity [88]. Upon binding to IFNAR1 and IFNAR2, IFN initiates a cascade that recruits and phosphorylates tyrosine kinase 2 (TYK2) and JAK1, leading to the activation of STAT1 and STAT2 [89,90]. The phosphorylated STAT1/STAT2 heterodimer binds to IRF9 in the cytoplasm, forming the IFN-stimulated gene factor 3 (ISGF3) complex, which translocates to the nucleus. There, it binds to ISG promoters, driving the transcription of hundreds of ISGs [91]. Viral inhibition of the JAK–STAT pathway is a common immune evasion strategy, as it prevents ISG production and IFN amplification. PRV inhibits IFN-induced STAT phosphorylation and ISG transcription [92]. Multiple studies have demonstrated that PRV disrupts the JAK–STAT pathway through mechanisms such as the degradation of key signaling molecules (such as IFNAR1 and JAKs), leading to the blockade of ISGF3 from binding to ISG promoters [93,94,95,96].

### 3.1. ISGs and the Antiviral Response

During PRV infection, ISGs can be induced through both interferon-dependent and interferon-independent mechanisms, helping the host establish an antiviral state. In response, PRV has evolved mechanisms to target ISGs and evade immune detection.

#### 3.1.1. OASL

Oligoadenylate synthase (OAS) is one of the key IFN effectors that act in the early stages of viral infection by degrading viral RNA, thereby inhibiting viral replication [97]. The OAS family consists of four distinct OAS isoforms, one of which is oligoadenylate synthase-like (OASL). Research by Chen et al. demonstrated that OASL restricts PRV replication by enhancing RIG-I signaling and increasing RIG-I-mediated IFN expression. Notably, PRV counters this antiviral response via the UL24 protein to suppress OASL transcription, thereby impairing the RIG-I signaling pathway and counteracting OASL antiviral activity [45] (Figure 4).

#### 3.1.2. ISG15 and ISG20

ISG15 and ISG20 are also critical in antiviral defense against PRV. ISG20 enhances IFN-β expression, but PRV UL24 suppresses ISG20 transcription, diminishing its antiviral effects [98]. Our studies indicate that ISG15 expression is upregulated during the early stages of PRV infection, where it impedes viral replication by increasing IFN-β expression and activating ISRE promoters [99]. Furthermore, ISG15 enhances IFN-α-mediated antiviral activity by facilitating STAT1/STAT2 nuclear translocation and STAT1/STAT2/IRF9 (ISGF3) complex formation [100]. Mice deficient in ISG15 presented increased susceptibility to PRV infection, although the mechanisms by which PRV evades ISG15-related antiviral immune responses remain to be elucidated [101]. Zhang et al. reported that PRV UL42 competitively binds to ISRE, thereby interfering with the binding of ISGF3 to ISRE, which results in decreased production of ISGs [94]. Additionally, the proteasomal degradation pathway is responsible for the degradation of Bclaf1 mediated by PRV US3, further impeding the binding of ISGF3 to ISRE [95] (Figure 4).

### 3.2. IFITM1 and IFITM2

IFN-inducible transmembrane proteins (IFITMs) are key host restriction factors with broad antiviral activity. IFITM2 plays a crucial role in controlling PRV by interfering with viral binding and entry [102]. Similarly, IFITM1 has significant antiviral functions, as PRV infection increases viral titers in IFITM1-knockdown cells and reduces PRV entry in IFITM1-overexpressing cells [103]. However, it is still unknown whether PRV antagonizes the IFITM function. A recently identified ISG protein, TMEM41B, promotes PRV replication by regulating lipid homeostasis [104].

### 3.3. PKR

Protein kinase R (PKR), which is activated by double-stranded RNA, is another ISG that plays a critical role in viral defense by phosphorylating eIF2α, reducing viral protein translation [105]. PRV has evolved mechanisms to counteract PKR activation and eIF2α phosphorylation [106,107]. For example, PRV suppresses eIF2α phosphorylation in vitro and in vivo, aiding in viral replication [108]. The PRV immediate-early protein IE180 also inhibits eIF2 phosphorylation during the early stages of viral replication [109]. While PKR is activated during PRV infection, the virus dephosphorylates eIF2α early on to increase virus replication [110].

### 3.4. TRIM26

Tripartite motif (TRIM) family proteins are characterized by a RING domain, B-box domains, a coiled-coil domain, and a variable C-terminal region that is responsible for interacting with diverse targets. These proteins play crucial roles in innate immunity, cell proliferation, autophagy, antiviral therapy, and tumor development. Among them, TRIM26 has been implicated in various biological processes, especially in regulating viral infections [111,112,113]. Wu et al. explored the interaction between TRIM26 and PRV and revealed that TRIM26 promotes PRV infection by facilitating NDP52-mediated autophagic degradation of MAVS. This discovery reveals a novel mechanism through which PRV evades host antiviral innate immunity, offering valuable insights into the interplay among viral infection, autophagy, and the innate immune response [114].

## 4. PRV Evades Apoptosis/Necrosis, Autophagy, and ER Stress

Upon PRV infection, viral proteins or nucleic acids trigger cellular stress responses, including apoptosis, necrosis, autophagy, and ER stress. These responses are critical for clearing the virus and maintaining cellular homeostasis. However, PRV, which can cause lifelong infections and severe diseases in various hosts, has developed multiple strategies to counteract these defenses.

### 4.1. Apoptosis

Apoptosis, also known as programmed cell death, is a vital defense mechanism against invading pathogens, including viruses. To evade this process, viruses have evolved sophisticated strategies to subvert apoptosis for immune evasion and enhanced viral replication. Several studies have shown that PRV infection induces apoptosis by upregulating the expression of pro-apoptotic Bcl family proteins [115], causing oxidative stress, free radical damage, and DNA damage [116]. Yeh et al. reported that PRV induces apoptosis via activation of the p38 MAPK and JNK/SAPK signaling pathways [117]. Additionally, low doses of Houttuynia cordata Thunb or emodin have been shown to inhibit PRV-induced apoptosis, reduce viral replication, and improve survival rates in mice [118,119].

Apoptosis is an important process regulating the pathogenesis of virus infection. Targeting apoptotic signaling during the late stages of infection represents a therapeutic strategy for herpesvirus-related diseases. Recent research revealed that caspases 3, 7, and 9 are activated in PRV-infected HeLa cells and that treatment with caspase inhibitors partially reduces apoptosis. PRV also modestly activates caspase-8 in PAMs and THP-1 cells, suggesting that caspase-8 may play a role in herpesvirus-induced apoptosis [120]. In addition to MAPK activation, PRV-induced oxidative stress and free radical formation are important contributors to apoptosis [117,121]. Interestingly, Aleman et al. reported that while PRV-infected neurons showed pathological changes, they presented no morphological or histochemical signs of apoptosis. Instead, apoptosis occurs primarily in the infiltrating immune cells surrounding PRV-infected neurons, suggesting that PRV may inhibit neuronal apoptosis to evade immune responses during acute infection [120].

PRV has developed several strategies to inhibit apoptotic signaling pathways. Notably, the PRV US3 protein has been shown to inhibit apoptosis in swine-testicle (ST) cells during the late stage of infection, depending on its enzyme activity [121]. Additionally, PRV activates the Akt and NF–κB pathways to protect infected cells from virus-induced apoptosis [121]. PRV US3 localizes to mitochondria and plays a crucial role in preventing the apoptosis induced by PRV [122]. Consistent with these findings, a recombinant virus lacking the US3 gene (PRV-∆US3) induces greater levels of apoptosis and has lower viral titers than wild-type PRV [123]. Broad-spectrum caspase inhibitors effectively inhibit apoptosis in ST and Hep-2 cells infected with either PRV-∆US3 or WT PRV [123]. A recent study elucidated the molecular mechanism by which the PRV glycoprotein M (gM), a late envelope glycoprotein, induces apoptosis during the late stage of infection. PRV-gM enhances PRV replication and pathogenicity by competitively interacting with BCL-XL to promote BAK oligomerization, ultimately activating caspase-3/7 [124].

### 4.2. Autophagy

Autophagy is an evolutionarily conserved catabolic process that plays a critical role in antiviral defense by degrading viral components and limiting viral replication. It is now recognized as an essential component of both innate and adaptive immunity. However, several herpesviruses, including PRV, have developed strategies to evade or exploit the autophagic machinery to increase their survival and replication [125,126]. Upon viral infection, host cells activate autophagy to degrade viral particles and components while simultaneously triggering an antiviral interferon response to antagonize viral replication.

Sun et al. explored the relationship between PRV replication and autophagy and reported that PRV induces autophagy in the early stages of infection [127]. However, as PRV infection progresses, PRV proteins suppress autophagy, increasing viral titers. Mechanistically, the tegument protein US3 may decrease autophagy levels by activating the AKT/mTOR pathway in infected cells [127]. These findings suggest that while autophagy initially contributes to PRV clearance, the virus has evolved mechanisms to antagonize this response. Xu et al. reported that PRV-induced autophagy enhances replication through the classical Beclin1–ATG7–ATG5 signaling pathway in neurons and that PRV infection promotes light chain 3 (LC3-I) autophagy in N2a cells [128]. Thus, PRV-induced autophagy both supports viral replication and antagonizes the host immune response.

Lyu et al. identified a regulatory role for Bcl2-associated athanogene 3 (BAG3), a chaperone-mediated selective autophagy protein that downregulates PRV lytic infection [129]. In light of the complex relationship between autophagy and PRV, several therapeutic targets have been identified. Xing et al. reported that Platycodon grandiflorus polysaccharide (PGPS) could inhibit PRV replication by upregulating the AKT/mTOR pathway, thus reducing PRV-induced autophagy [130]. Ming et al. demonstrated that inhibiting deubiquitinase 14 (USP14) promotes the ubiquitination of PRV VP16, facilitating its degradation via SQSTM1/p62-mediated selective autophagy. Pretreatment with the USP14 inhibitor b-AP15 activates ER stress and autophagy, inhibiting PRV infection in vivo [131] (Figure 5), suggesting that USP14 is a potential therapeutic target for treating alpha-herpesvirus infections.

Ma et al. described a distinct mechanism by which PRV exploits TOLLIP-mediated selective autophagy to circumvent host immunity [64]. Their study revealed that the PRV tegument protein UL21 inhibits type I IFN signaling by degrading cGAS through the autophagic degradation of STING by recruiting TOLLIP, thereby inhibiting cGAS–STING antiviral signaling [66] (Figure 5), representing a significant immune evasion strategy employed by the PRV. Wu et al. also demonstrated that PRV-induced TRIM26 degrades MAVS through NDP52-mediated selective autophagy, providing further insight into the virus’s ability to escape antiviral innate immunity and the complex interactions among PRV, autophagy, and the host immune response [114].

### 4.3. ER Stress

The endoplasmic reticulum (ER) is a vital organelle responsible for key cellular processes, including protein synthesis, folding, transport, and secretion. The ER can detect various external stimuli, such as pathogen invasion, heat shock, hypoxia, and the accumulation of unfolded or misfolded proteins. In response, the ER activates the unfolded protein response (UPR) to restore cellular homeostasis [132]. Viral infections can also trigger autophagy via the UPR pathway [133,134,135].

Several studies have shown that viral infections induce ER stress, and some viruses exploit host cell membranes to build membrane components for their progeny virions during this process [136]. Alpha-herpesviruses, including PRV, have been found to suppress ER stress to increase viral replication. Yang et al. demonstrated that glucose-regulated protein 78 (GRP78), a marker of ER stress, is upregulated in the early stages of PRV infection, suggesting that PRV induces ER stress and activates UPR pathways [137]. Recent research has provided new insights into PRV replication and pathogenesis, showing that the activation of protein kinase RNA-like ER kinase (PERK) and inositol-requiring protein-1 (IRE1) pathways supports PRV replication in BHK21 cells, with glycoprotein B playing a pivotal role in inducing ER stress [138]. These findings expand our understanding of ER stress and the role of UPR in viral replication.

## 5. Conclusions

PRV can establish lifelong infections in hosts, leading to severe complications, especially in immunocompromised individuals. It is also a common cause of congenital viral infections. PRV triggers several host defense mechanisms, with the innate immune response being the first line of defense, primarily limiting viral infection by producing type I IFN and ISGs. Throughout the ongoing battle between viruses and hosts, PRV has developed various strategies to evade immune defenses, facilitating its survival and replication. Multiple viral proteins, particularly envelope and IE proteins, target key adapters in antiviral signaling pathways such as TLR2/4, cGAS, STING, NF–kB, TBK1, and IRF3 to inhibit host immune processes, including apoptosis, necrosis, autophagy, and ER stress, contributing to immune evasion and persistent infection.

Emerging evidence highlights the potential of TLR agonists as antibacterial, anticancer, and vaccine adjuvants, as well as their importance in vaccine development and immunotherapy against viruses such as HSV, HBV, and HIV. TLR antagonists have also been explored as therapeutic agents to suppress excessive immune responses. This review of the immune evasion mechanisms of PRV, particularly its ability to target PRRs and cellular processes, may offer valuable insights for the development of antiviral drugs and vaccines.

## 6. Future Perspective

As a member of the herpesvirus family, PRV serves as a valuable model for investigating the interactions between viral and host cellular mechanisms, particularly in the context of modulating innate immune signaling pathways and altering the antiviral state of host cells. Given its broad host range and the functional similarities of its encoded proteins to those of HSV, PRV offers significant potential for elucidating the roles of alpha-herpesvirus proteins in various physiological processes within host cells. This understanding may provide critical insights for the development of novel antiviral strategies. Based on the study of virus–host interactions, novel drug targets can be identified, facilitating the development of innovative antiviral therapeutics. Furthermore, our observations have highlighted the significant effects of PRV on cell death, including apoptosis, necrosis, and autophagy, as well as the interplay between viral infection and ER stress-related responses. This review elucidates the diversity of cellular antiviral mechanisms and broadens the scope for future research in antiviral immunity.

## Figures and Tables

**Figure 1 ijms-25-13140-f001:**
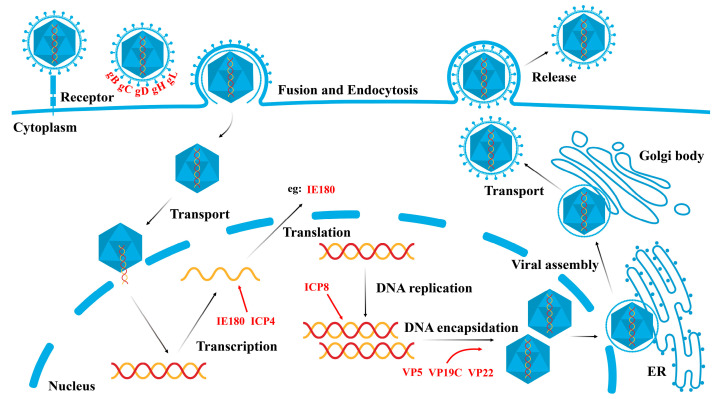
Life cycle of a PRV. The life cycle of PRV begins with the interaction of PRV glycoproteins with specific receptors on the host cell surface, triggering endocytosis and fusion with the cell membrane. Once inside the cell, the viral nucleocapsid is transported to the nucleus, where the viral DNA is released. Inside the nucleus, viral genes undergo transcription, and the viral genome is subsequently replicated. The viral proteins are translated in the cytoplasm and transported back to the nucleus to participate in the assembly of the nucleocapsid. The nucleocapsid then exits the nucleus and is transported to the endoplasmic reticulum (ER) and Golgi apparatus, where it associates with membrane proteins and glycoproteins to form mature viral particles. These mature viral particles are ultimately released into the extracellular space through the process of budding.

**Figure 2 ijms-25-13140-f002:**
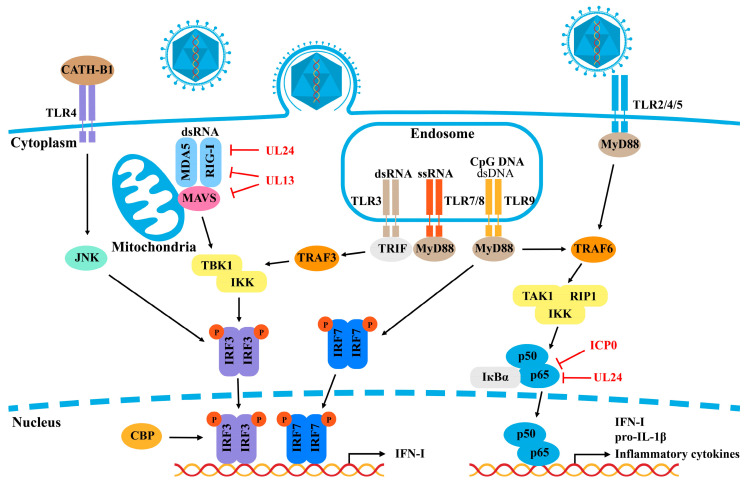
PRV evades the TLR and RLR signaling pathways. TLRs, which are located on the cell membrane and within endosomes, detect viral components, such as ssRNA, dsRNA, dsDNA, and CpG DNA. Upon recognition, TLRs transmit signals through TRIF and MyD88 adaptors, leading to the activation of IRFs and NF–κB. TLRs can also detect peptides such as cathelicidins, further activating IRF3 and inducing IFN-β production. In the RLR signaling pathway, RIG-I and MDA5 sense viral RNA and signal through MAVS on the outer mitochondrial membrane, triggering the activation of IRFs and NF–κB. Both the TLR and RLR pathways produce interferons and inflammatory factors, which contribute to the antiviral response. PRV-encoded proteins inhibit the TLR and RLR signaling pathways, thereby allowing the virus to evade immune detection and response.

**Figure 3 ijms-25-13140-f003:**
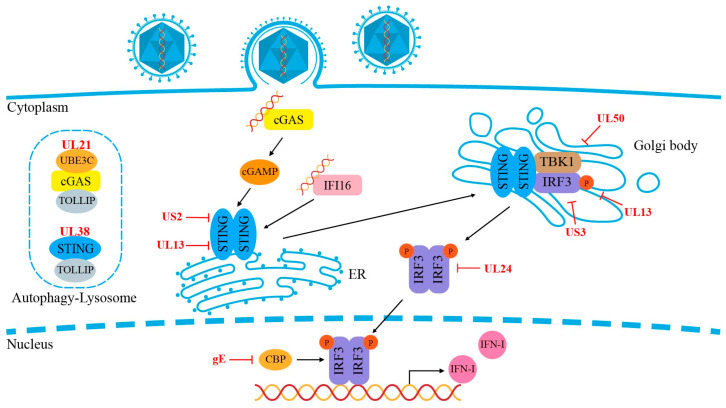
PRV evades the DNA-sensing signaling pathway. Cytoplasmic DNA sensors, such as cGAS and IFI16, recognize viral double-stranded DNA and transmit signals to IRF3, resulting in the production of IFNs. PRV-encoded proteins can target multiple steps within the DNA sensing signaling pathway, effectively inhibiting signal transduction and facilitating immune evasion.

**Figure 4 ijms-25-13140-f004:**
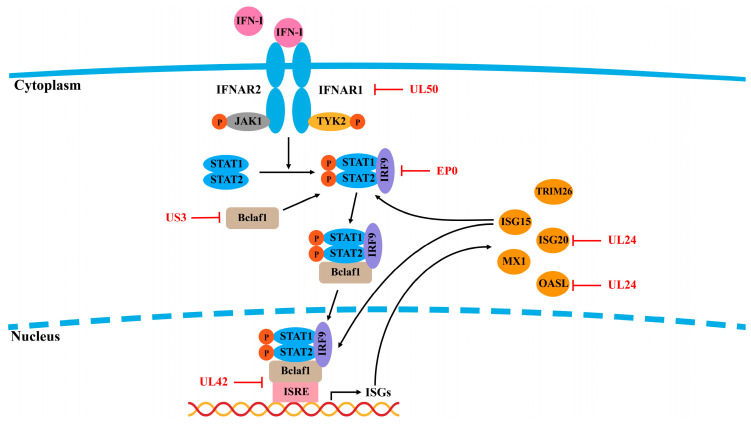
PRV evades the IFNAR–JAK–STAT signaling pathway. Type I IFNs exert their antiviral effects by binding to their heterodimeric receptors, IFNAR1 and IFNAR2, thereby activating the downstream JAK–STAT signaling pathway, which promotes the production of ISGs. Some ISGs, such as ISG15, can also enhance the JAK–STAT signaling pathway to upregulate the production of interferon-stimulated factors and increase antiviral activity. PRV-encoded proteins target components of the IFNAR–JAK–STAT signaling pathway to inhibit signal transduction or directly interact with ISGs, facilitating immune evasion.

**Figure 5 ijms-25-13140-f005:**
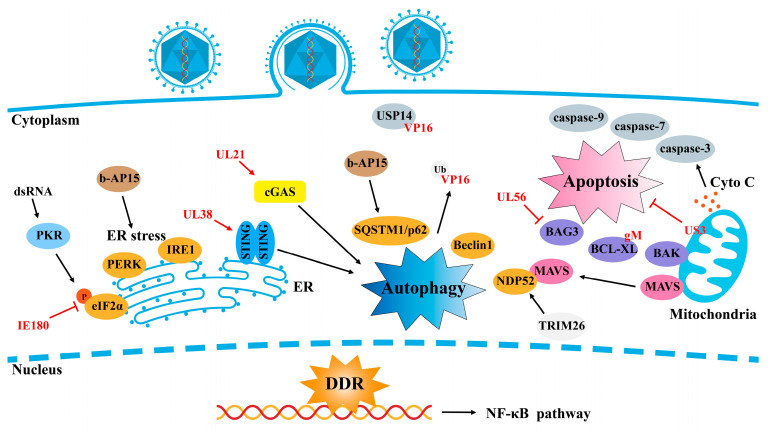
PRV evades apoptosis/necrosis, autophagy, and ER stress. During PRV infection, cells can undergo apoptosis, necroptosis, autophagy, and ER stress in response to viral nucleic acids and proteins, which are aimed at limiting viral infection and exerting antiviral effects. However, PRV-encoded proteins can counteract these responses or target host proteins for autophagic degradation, thereby facilitating the evasion of the host immune response by PRV.

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
