# Peer review of "Evasion of the Antiviral Innate Immunity by PRV"

_ijms, 2024, doi:10.3390/ijms252313140_

Round 1
Reviewer 1 Report
Comments and Suggestions for Authors
Estimated Authors,
I had the pleasure to read this wholesome but also concise review on the topic of pseudorabies virus cellular biology. Authors have done a very good job in summarizing and systematizing all we currently know about this specific virus and how it interacts with cellular pathways.
The potential significance of this review is clearly recognizable, as it could provide some significant insights on the biology of other alphaherpesviruses.
Briefly, the paper has been well written, and I've no specific recommendations of complaints about the design and development of this review.
For the present reviewer, the paper could be accepted as it is.
Author Response
Thanks very much for your recognition of our work.
Reviewer 2 Report
Comments and Suggestions for Authors
Wang et al. have written a comprehensive, clearly organized and informative survey of how pseudorabies virus, an alpha-herpesvirus, evades antiviral innate immune responses. A few errors must be corrected, and some statements should be clarified, but after these revisions the manuscript will merit publication.
1. Line 54. This statement is incorrect: translation does not occur in the nucleus. Consequently, the scheme in this figure is incorrect and also over-simplified, because viral proteins synthesized in the cytoplasm must be imported into the nucleus before being assembled into capsids and re-exported.
2. Line 364. This statement is incorrect and must be corrected. PRV does not phosphorylate eIF2, and the cited paper clearly states that " PRV encoded-functions dephosphorylate eIF2α".
3. Line 341. When printed in black and white, it is difficult to see the names of viral proteins against the colored background.
4. Line 391. What are " Houttuynia cordata Thunb" and "emodin"?
5. Grammatical and related errors
a) Lines 48-9. The meaning of this sentence should be clarified. Do the authors really mean that sensory neurons remain dormant (true, but not obviously relevant to the topic of this review), or that PRV remains dormant?
b) Line 87. The authors presumably mean "to promote" rather than "from promoting".
c) Line 128. The host-derived RNAs are generated by the host, and not by viruses.
d) Lines 178, 180, 266. Please explain the acronym "GSDMD".
e) Lines 249-251. This sentence should be rewritten, because it reads as if STING is involved in viral immune evasion, whereas PRV US2 is presumably intended.
f) line 349. Please explain the acronym " Bclaf1 ".
g) line 398. Please explain the acronym " PAMs".
h) The problem with acronyms is pervasive - many are not defined, which limits comprehensibility for non-specialists. Please correct.
6. References
a. Lines 243-4. In addition to references 19 and 74, it would be appropriate to cite an earlier paper: Bo et al. (2020). PRV UL13 inhibits cGAS-STING-mediated IFN-β production by phosphorylating IRF3. Vet Res. 5:118.
b. Lines 494, 495. The authors cite papers that don't appear in the reference list.
c. Lines 506-7 and 521-2, and lines 526-7 and 618-9: these references are duplicated. Given that repetitions occurred twice, the authors should check the entire reference list for possible further duplications.
Comments on the Quality of English Language
See sub-section 5 in my comments.
Author Response
Line 54. This statement is incorrect: translation does not occur in the nucleus. Consequently, the scheme in this figure is incorrect and also over-simplified, because viral proteins synthesized in the cytoplasm must be imported into the nucleus before being assembled into capsids and re-exported.
Re: Thanks for your constructive suggestion. We have made changes to the text and figures to address the suggested revisions, and also added relevant viral proteins involved in viral genome replication, transcription, and nucleocapsid synthesis.
Line 364. This statement is incorrect and must be corrected. PRV does not phosphorylate eIF2, and the cited paper clearly states that " PRV encoded-functions dephosphorylate eIF2α".
Re: Thanks for your advice and we have corrected it in the revised manuscript.
Line 341. When printed in black and white, it is difficult to see the names of viral proteins against the colored background.
Re: Thank you very much for your suggestion. We have removed the red background and change the font color of the viral protein names to red.
Line 391. What are " Houttuynia cordata Thunb" and "emodin"?
Re: Houttuynia cordata Thunb, commonly known as Houttuynia or fish mint, is a perennial plant native to Southeast Asia. It is part of the Saururaceae family and is often used in traditional medicine and culinary practices, especially in countries like China, Japan, and Korea. The plant is known for its distinctive heart-shaped leaves and has been used in herbal remedies for its purported anti-inflammatory, antimicrobial, and antioxidant properties. It's also sometimes used in salads, soups, and other dishes for its unique flavor.
Emodin is a chemical compound that is a type of anthraquinone. It is found in several plants, including Rheum (rhubarb), Polygonum cuspidatum (Japanese knotweed), and Houttuynia cordata. Emodin has been studied for its potential pharmacological properties, such as anti-inflammatory, antioxidant, and anticancer effects. Some research has suggested that emodin may also have benefits in regulating blood sugar levels, promoting liver health, and even inhibiting the growth of certain bacteria or fungi.
Grammatical and related errors
a) Lines 48-9. The meaning of this sentence should be clarified. Do the authors really mean that sensory neurons remain dormant (true, but not obviously relevant to the topic of this review), or that PRV remains dormant?
Re: Thanks for your advice. Our mean is that PRV remains dormant, and this has been revised in the manuscript.
b) Line 87. The authors presumably mean "to promote" rather than "from promoting".
Re: We agree with your advice and have revised it.
c) Line 128. The host-derived RNAs are generated by the host, and not by viruses.
Re: We agree with your advice and have revised it.
d) Lines 178, 180, 266. Please explain the acronym "GSDMD".
Re: GSDMD (Gasdermin D) is a protein that plays a key role in the process of pyroptosis, which is a form of programmed cell death associated with inflammation. It is part of the gasdermin family of proteins, which have been implicated in the regulation of cell death and inflammation.
e) Lines 249-251. This sentence should be rewritten, because it reads as if STING is involved in viral immune evasion, whereas PRV US2 is presumably intended.
Re: We agree with your advice and have revised in the manuscript.
f) line 349. Please explain the acronym " Bclaf1 ".
Re: Bclaf1 stands for Bcl-2-associated transcription factor 1. It is a protein that plays a role in regulating apoptosis (programmed cell death), cell survival, and transcription.
g) line 398. Please explain the acronym " PAMs".
Re: PAMs refer to perivascular macrophages (sometimes called PAM cells), which are a specific type of immune cell found in tissues surrounding blood vessels, particularly in the brain and central nervous system (CNS). These cells are a subset of macrophages, which are important for immune defense and tissue maintenance.
h) The problem with acronyms is pervasive - many are not defined, which limits comprehensibility for non-specialists. Please correct.
Re: We agree with your advice and have revised it in the revised manuscript.
- References
- Lines 243-4. In addition to references 19 and 74, it would be appropriate to cite an earlier paper: Bo et al. (2020). PRV UL13 inhibits cGAS-STING-mediated IFN-β production by phosphorylating IRF3. Vet Res. 5:118.
Re: Thanks for your advice and we accept it.
- Lines 494, 495. The authors cite papers that don't appear in the reference list.
Re: Thanks for your comment. We have revised it in the manuscript.
- Lines 506-7 and 521-2, and lines 526-7 and 618-9: these references are duplicated. Given that repetitions occurred twice, the authors should check the entire reference list for possible further duplications.
Re: Thank you for your suggestion. We have carefully checked the references and removed the repeated references.

Reviewer 3 Report
Comments and Suggestions for Authors
The content of this article is quite rich, and the author provides an in-depth introduction of the content of the immune evasion strategies of PRV. However, the article also has some shortcomings, and specific suggestions are as follows:
The full text only contains objective comments and conclusions, but lacks expectations for future development and personal summaries.
This article elaborates on existing research results in detail, but lacks sufficient description of the problems and directions that need to be further explored.
Line 169-180:The sudden mention of AIM2 at the beginning of this paragraph seems a bit abrupt and unrelated to the previous paragraph. It is recommended to add some linking statements
In section 2.4, it is mentioned that the receptors of the DNA sensing signaling pathway also include cGAS, STING, DDX41, DHX9, IFI16, and ZBP1. Subsequent sections 2.5, 2.6, and 2.7 all discuss the DNA sensing signaling pathway. Since they are all receptors of the DNA sensing signaling pathway, why not classify them under section 2.4.
Line 314-317:The content of 3.1 is too limited and there are no references marked.
Line 383-418:In section 4.1, it is mentioned that PRV induced cell apoptosis is a necessary condition for pathogenesis, and it is also mentioned that cell apoptosis is a defense mechanism of the body, which seems a bit confusing. It is suggested to combine these two points for comments and explanations at the end.
Author Response
This article elaborates on existing research results in detail, but lacks sufficient description of the problems and directions that need to be further explored.
Re: Thank you for your advice. We have supplemented the future research development of PRV in the manuscript
Line 169-180: The sudden mention of AIM2 at the beginning of this paragraph seems a bit abrupt and unrelated to the previous paragraph. It is recommended to add some linking statements.
Re: Thanks for your advice and we accept it.
In section 2.4, it is mentioned that the receptors of the DNA sensing signaling pathway also include cGAS, STING, DDX41, DHX9, IFI16, and ZBP1. Subsequent sections 2.5, 2.6, and 2.7 all discuss the DNA sensing signaling pathway. Since they are all receptors of the DNA sensing signaling pathway, why not classify them under section 2.4.
Re: Thank you for pointing it out. The title setting problem has been modified in the revised manuscript
Line 314-317: The content of 3.1 is too limited and there are no references marked.
Re: The issue with the title setting has also been addressed and modified accordingly.
Line 383-418: In section 4.1, it is mentioned that PRV induced cell apoptosis is a necessary condition for pathogenesis, and it is also mentioned that cell apoptosis is a defense mechanism of the body, which seems a bit confusing. It is suggested to combine these two points for comments and explanations at the end.
Re: Thanks very much for your suggestive advice. We are sorry for the lack of precision in our statement about “PRV-induced apoptosis is essential for its pathogenicity”. We have revised it in the revised manuscript as follows: Apoptosis is an important process regulating the pathogenesis of virus infection.
